# Long-Term Psychological Effects of COVID-19 Pandemic on Children in Jordan

**DOI:** 10.3390/ijerph18157795

**Published:** 2021-07-22

**Authors:** Harran Al-Rahamneh, Lubna Arafa, Anas Al Orani, Rahaf Baqleh

**Affiliations:** School of Sport Sciences, The University of Jordan, Amman 11942, Jordan; lubnaarafeh90@yahoo.com (L.A.); Anas.sport100@yahoo.com (A.A.); rahafbaqleh98@outlook.com (R.B.)

**Keywords:** COVID-19, psychological impacts, 5–11 year old children, school closure, screen time usage, physical activity

## Abstract

Millions of children and adolescents have been affected worldwide by quarantine, school closures, and social distancing measures which have been implemented by many countries to control the spread of COVID-19. However, the long-term consequences of such procedures on children’s well-being are not clear. Therefore, this study investigated the psychological impacts of COVID-19 on Jordanian children between the ages of 5–11 years old. A total of 1309 parents with children between the ages of 5 and 11 years old filled in an online survey that included a set of questions to measure their children’s behaviour and emotions before and during the COVID-19 pandemic. Being bored (77.5%), irritable (66%), likely to argue with the rest of the family (60.7%), nervous (54.8%), reluctant (54.2%), and lonely (52.4%) were the most frequently reported symptoms compared to the pre-COVID-19 period. Parents reported that screen use of ≥120 min a day was shown among 48.9% of children and 42% of children did <30 min a day of physical activity. ≤8 h of sleep per night was reported among 42.5% of children compared to pre COVID-19. The results emphasized the importance of developing preventative psychological programs to minimize the impact of the COVID-19 pandemic on children’s psychological well-being.

## 1. Introduction

In December 2019, the first cases of a novel pneumonia were reported in Wuhan, the capital of Hubei province in China [1]. The virus then spread quickly among many countries around the world. At the beginning of January 2020, the novel coronavirus was linked to the severe acute respiratory disease called COVID-19 [2]. The World Health Organization (WHO) declared it as a pandemic on 11 March 2020 due to its serious and rapid spread [2]. To date, there have been 186,638,285 people infected by COVID 19 and 4,035,037 have died as a result of the infection worldwide [3]. In Jordan, there have been approximately 757,690 people reported to be infected by COVID 19, and 9843 deaths [4]. 

The COVID-19 pandemic has severely changed the daily lives of all individuals in a very short period worldwide. Limited travel, imposed quarantines, lockdowns, suspended public gatherings, and closed businesses as well as universities and schools are only some of the policies and measures that most countries have adopted to limit the spread and mitigate the negative health outcomes of the virus [5]. However, these policy measures have affected most sectors. For example, prices of agricultural commodities dropped by 20% as a consequence of less demand from hotels and restaurants [6]. High healthcare costs, shortages of protective equipment including N95 face masks, and low numbers of ICU beds and ventilators have ultimately exposed weaknesses in the delivery of patient care [7]. In addition, in the USA, the unemployment rate rose from 3.8% in February 2020 to 14.7% in April 2020, with 23.1 million unemployed [8].

The education system was also affected. Individuals with a higher income can access technology that can ensure education continues digitally during social isolation, whereas some lower income families cannot. In Dubai, for example, 13,900 people have signed an appeal to decrease independent school fees by 30% as parents struggle to source these funds amidst recent pay cuts reaching as high as 50%, and high costs of living [7]. The long-term effect of school closure has not been studied yet. However, Chen et al. [9] reported that 27% of families could not go to work, with 18% losing income as a direct result of a one-week closure of schools in Taiwan during the outbreak of H1N1 in 2009.

Brooks et al. [10] reported that the lockdown is associated with poor social and emotional well-being in adults. Children and adolescents may be more susceptible because of home confinement, school closure, lack of in-person contact with classmates, friends and teachers, and limitation in personal space at home [10,11,12,13,14]. Orgilés et al. [14] showed that more than 85% of parents in Spain and Italy reported changes in their children’s emotional state and behaviours; for example (76.6%) of children had difficulties in concentration, and (52.1%) of children felt more bored. In addition, Morgül et al. [15] reported that children felt more bored (73.8%), lonely (64.5%), and frustrated (61.4%) during lockdown compared to pre-lockdown in the UK. These changes in children’s well-being and emotional state may be attributed to less physical activity and increases in domestic violence as a result of house confinement [16]. In addition, these changes are speculated to be affected by parents’ stress [12]. For example, Spinelli et al. [17] found that parents who reported more difficulties in dealing with quarantine were more stressed which, in turn, increased children’s emotional and behavioural difficulties.

The first case of COVID-19 occurred in Jordan in March 2020 and based on that full lockdown was implemented by the Jordanian government until the end of May 2020. All schools were closed and face-to-face teaching was replaced by e-learning for more than one year (March 2020–present). This may affect children’s behaviour and emotional state. To the best of our knowledge, there are no published studies in Jordan which have investigated the long-term impact of the COVID-19 pandemic on children’s behaviour, emotional state, and well-being. Therefore, this study was conducted to assess the effect of COVID-19 pandemic on psychological well-being among children between the ages of 5–11 years old in Jordan. This study will have important implications for ministries of education and health in Jordan by adding some lectures on the official teaching platform on how parents can manage their children’s stress and mental health during such difficult times. In addition, these ministries should provide more lectures on the importance of increasing physical activity and decreasing screen time usage for children. We hypothesized that the COVID-19 will negatively affect the psychological well-being of Jordanian children between the ages of 5–11 years old. We also hypothesised that this negative effect of COVID-19 on the psychological well-being of Jordanian children will be affected by parents’ gender and social status and children’s age. Finally, we hypothesised that physical activity level will be decreased and screen time usage will be increased among the study sample due to COVID-19.

## 2. Materials and Methods

### 2.1. Participants

Data for the current study was collected from 1309 parents (mothers = 1219 (93.1%); fathers = 90 (6.9%) on the long-term psychological effect of COVID-19 on children’s emotional and behavioural state. As shown in Table 1, most parents lived in cities (97.2%) and the rest lived in a village (2.3%), camp (0.2), or Bedouin (0.3). Additionally, most participants were married (94.3%), and had a full-time job (79.5%). Most of the parents had completed a bachelor degree or postgraduate studies (mothers = 982 at 75% and fathers = 908 at 69%). Fathers were more likely to have a full-time job than mothers (mothers = 403 at 31% and fathers = 638 at 49%. Unemployment was higher among mothers (*n* = 600, 45.8%) than fathers (*n* = 39, 3%). Most participants lived in an owned household (68.4%). Half of the participants (50.9%) had balconies and the rest of the participants (49.1%) had access to outside space for their children to play or hang out, which in most cases was a garden (34.8%). Regarding children (*n* = 1309 male = 716 (54.7%) and female = 593 (45.3%)), they were between 5–11 years of age (8.1 ± 2.02 years) and most of them were studying at private schools (92.7%).

### 2.2. Procedures

The survey was uploaded and shared on the Google online survey platform. A link to the electronic survey was distributed via social networks (e.g., Facebook, Instagram), e-mails, and messaging groups (e.g., WhatsApp) during the period 10 April 2021–17 April 2021 using a snowball sampling strategy, since face-to-face contact was not possible for all individuals due to the COVID-19 pandemic. The survey was distributed to parents’ pages and only parents who live in Jordan were asked to fill in the survey. Information about the objectives of the study was provided and informed consent was requested. Parents were also informed that they will not be paid for filling in the survey. Institutional ethics approval was obtained by the school of Sport Sciences at the University of Jordan.

### 2.3. Instrument

The children’s emotional and behavioural symptoms questionnaire was developed by Orgilés et al. [14] to assess children’s mental health during the COVID-19 lockdown. This includes (a) parents and child sociodemographic information including housing conditions (e.g., house size, number of rooms, the existence of outside space such as garden, balcony, or terrace); (b) parents’ perceived impact of the confinement on children’s emotional and behavioural symptoms rated on a five-point scale (1 = much less compared to before lockdown–5 = much more compared to before lockdown); (c) children’s daily routines during lockdown compared to before: screen time usage and duration of physical activity rated on a six-point scale (1 = less than 30 min–6 = more than 180 min), and sleep hours a day. Orgilés et al. [14] questionnaire was translated to the Arabic language by Dr. Harran Al-Rahamneh and the Arabic version was checked by three of the academic staff members at the school of Sport Sciences at the University of Jordan and was translated back to English by Rahaf Baqleh to ensure the accuracy of the translation. The questionnaire demonstrated excellent internal consistency and reliability (Cronbach’s alpha = 0.907).

### 2.4. Data Analysis

Statistical analyses were conducted using the IBM SPSS (Statistical Package for the Social Sciences) software version 16.0. Mainly, means, standard deviation, and percentages were used. Spearman’s correlation coefficient was used to assess the association between parents’ perceived change in their children’s physical activity and children’s screen time usage. An independent sample t-test was used to compare whether there was a difference in the total score of children’s behaviour and emotional state during compared to before COVID-19 between boys and girls. In addition, a series of analyses of variance (ANOVA) was used to compare whether there was a difference in the mean score of children’s behaviour and emotional states during compared to before COVID-19 due to marital status, children’s age, available outside spaces and fathers and mothers’ educational and employment status. Leven’s test was used to check homogeneity of variance in t-test and ANOVA and if this assumption was violated, degrees of freedom were adjusted.

## 3. Results

In order to assess parents’ perceived changes in children’s behaviour and emotional state during compared to before COVID-19, the percentages of somewhat more and much more were summed and the percentage of much less and somewhat less were summed. For example, 31.3% of parents reported changes in “my child is worried” as somewhat more and 7.4% of them reported these changes as much more; therefore, the percentage of parent’s perceived changes in this item is 38.7%. Parents who noticed changes in their children’s behaviour and emotional states before and during the COVID-19 are presented in Table 2. According to parents’ reports (Table 2), children were more bored (77.5%), irritable (66%), more likely to argue with the rest of the family (60.7%), nervous (54.8%), reluctant (54.2%), lonely (52.4%), angry (51.8%), restless (48.6%), cries easily (47.3%), difficulty concentrating (46.1 %), anxious (44.8%), dependent on us (44.2%), sad (43.4%), uneasy (42.9%), frustrated (42.7%), worried (38.7%), and were afraid of COVID-19 infection (38.6%) during the lockdown compared to the pre-COVID-19 period. In addition, 51.5% of parents perceived their children to be less quiet and 43.8% of parents reported that their children have no appetite.

Independent sample t-test showed that there was no significant difference in children’s mean score of psychological behaviour and emotional state during compared to pre-COVID-19 between boys and girls t_(1307)_ = −1.756, *p* = 0.079.

ANOVA showed that there was no significant difference in children’s mean score of psychological behaviour and emotional state during compared to before COVID-19 between the categories of parents’ available outside spaces (garden, balcony, the roof of the building, basement) F_(3, 1305)_ = 0.525, *p* = 0.665, between the categories of mother’s educational level (less than high school, high school, diploma, bachelor, postgraduate) F_(4, 1304)_ = 0.905, *p* = 0.460, between the categories of father’s educational level (less than high school, high school, diploma, bachelor, postgraduate) F_(4, 1304)_ = 0.963, *p* = 0.427 and between the categories of mother’s employment status (self-employed, part-time, full-time, unemployed, retired, lost jobs due to covid-19) F_(5, 1303)_ = 1.393, *p* = 0.224.

However, ANOVA showed that there was a significant difference in children’s mean score of psychological behaviour and emotional state during compared to before COVID-19 between the categories of children’s age groups, F_(6, 1302)_ = 2.718, *p* = 0.013, between the categories of parent’s marital status (married, widowed, divorced, never married), F_(3, 1305)_ = 2.958, *p* = 0.031, and between the categories of father’s employment status (self-employed, part-time, full-time, unemployed, retired, lost jobs due to covid-19), F_(5, 1303)_ = 3.635, *p* = 0.003. Post hoc analysis using Tukey’s honestly significant difference showed that 9-year-old children were more affected than 11-year-olds (*p* = 0.041); children of divorced parents were more affected than those of single caregivers (*p* = 0.056) and those of married couples (*p* = 0.061); and children of parents who lost their jobs as a result of COVID-19 were more affected than those of retired parents (*p* = 0.011) and those of self-employed parents (*p* = 0.003). There was a negative relationship between physical activity and screen time usage (r = −0.175, *p* = 0.000). Screen time usage in minutes per day, physical activity in minutes per day and sleeping hours per day are presented in Table 3.

## 4. Discussion

This study assessed parents’ perceived changes in children’s behaviour and emotional state during compared to before COVID-19. The results showed that a high percentage of the parents noticed mild or great changes in their child’s emotional state and behaviour during the school closure. An increase in boredom of 77.5%, followed by an increase in irritability of 66% and being more likely to argue with the rest of the family of 60.7%, were the most prominent changes reported by the parents. Increases were also noted in nervousness, reluctance, loneliness, and anger by approximately half of the parents. Additionally, 43.8% of the parents reported that their children had no appetite. These findings are in agreement with Davico et al. [12]; Orgilés et al. [14], Morgül et al. [15], Sprang & Silman, [18]; Spinelli et al. [17], and Wang et al. [16]. Orgilés et al. [14] observed that difficulty in concentrating was the most frequent symptom, at 76.6%, reported by parents among children in Spain and Italy. In addition, these authors observed that boredom, irritability, restlessness, nervousness, feelings of loneliness, and being more uneasy and more worried were reported by more than 30% of parents. Moreover, these psychological symptoms were higher among children in Spain compared to Italy, which might be attributed to some measures applied by the governments. For example, the Italian government allowed children and youths under 18 years old to go for a short walk during the confinement which might, in turn, have minimized these psychological symptoms. Morgül et al. [15] showed that boredom, at 73.8%, and loneliness, at 64.5%, were the most frequent psychological symptoms reported by parents among children of the same age group in the UK. Sadness, frustration, irritability, restlessness, worries, and anger were reported by 50% of parents during the lockdown compared to the pre-COVID-19 period. In addition, Sprang and Silman [18] showed that the mean post-traumatic stress scores were four times higher in children who had been quarantined than in those who were not quarantined. This can be attributed to the fact that during school-time teachers have roles not only in delivering educational materials but also in offering an opportunity for children to interact, and to receive support and explanations [17]. In addition, Brazendale et al. [19] and Wang et al. [20] showed that children are physically less active, have much longer screen time usage, irregular sleep patterns, and less favourable diets, resulting in weight gain and a loss of cardiorespiratory fitness when children are in weekends and summer holidays. Moreover, Wang et al. [16] reported that lockdown due to COVID-19 restricted outdoor activities and interaction with same-aged friends during the outbreak among children, which in turn will worsen such negative effects. These perceived changes in physical activity, screen time usage, and sleeping hours observed by Spinelli et al. [17], Wang et al. [20], and Wang et al. [16] might in turn increase psychological symptoms among children such as boredom, irritability and likelihood to argue with the rest of the family during school closures due to COVID-19.

Our results showed that 42.0% of children performed less than 30 min of physical activity a day. In addition, screen time usage of 180 min or more among children was reported by 34.5% of parents. These findings are in agreement with previous studies. Screen time usage of more than 180 min a day was found among 29.9% of youth in Italy and Spain, and 55.6% of studied youths performed less than 30 min of physical activity a day during quarantine compared to before quarantine, as reported by Orgilés et al. [14]. Similarly, Morgül et al. [15] showed that for children between the ages of 5–11 years old in the UK, those with screen time usage of 180 min or more a day increased from 1.4% before the lockdown to 33.8% during the lockdown, and an increase was also shown in doing less than 30 min of physical activity a day, from 3.7% before compared to 16.2% during the lockdown. In addition, Abid et al. [21] reported an increase in screen time usage from 1.24 to 3.38 h a day among Tunisian children. Abid et al. [21] also showed a reduction in physical activity during compared to pre-lockdown among Tunisian children. In our study, there was a significant inverse relationship between physical activity and screen time usage. This can be attributed to the fact that children between the ages of 5–11 years old in Jordan had two physical education classes per week in addition to outbreak activities before school closure due to COVID-19. Therefore, school closure and full and partial lockdown due to COVID-19 led to less physical activity, more screen time usage, irregular sleep patterns, and less appetite.

Lockdown and school closure also affected sleeping hours. In our study, 42.5% of children had 8 h or less of sleep per night, which is considered as a short sleeping duration [22]. Morgül et al. [15] showed that children slept for half an hour more before than during the lockdown. In addition, Abid et al. [21] also showed a change in sleeping hours among Tunisian children. They reported a decrease in sleeping hours among boys by 0.07 h during (8.71 ± 0.93 h) compared to before lockdown (8.78 ± 0.95 h) and an increase in sleeping hours among girls by 0.08 h during (8.73 ± 0.78 h) compared to before lockdown (8.65 ± 0.72 h). These changes in sleeping hours may be attributed to less physical activity, more screen time usage, and changes in psychological behaviour and emotional state during the lockdown compared to before COVID-19.

Parent’s perceived changes in children’s behaviour and emotional state before and during the COVID-19 were not affected by children’s gender, parent’s gender, parent’s (fathers and mothers) educational level and parent’s available outside space. However, children of divorced parents were more affected than those of single caregivers. This is not a surprising finding since more stressed families due to difficulties in dealing with quarantine are related to increased children’s emotional and behavioural difficulties [17]. Similarly, Davico et al. [12] showed that children’s post-traumatic stress scores were related to their parents’ post-traumatic stress symptom scores and this may suggest a possible family effect for distress, especially when considering the correlation between siblings and single children. Maiti et al. [23] reported that the strength of a good marital or couple bonding can actually make both the individuals internally strong and confident, which definitely help to fight a given distress in a better way, individually and also as a strong united couple. Therefore, divorced couples usually do not have such a good marital relationship, which is going to be reflected negatively on their children. This may also be attributed to the fact that children of divorced families may have spent less time with caregivers if children were unable to travel from one parent’s residence to another during the lockdown, or if this transfer occurred less frequently.

Moreover, children of parents who lost their job due to COVID 19 were more affected than those of retired parents and those of self-employed. In Jordan, during the lockdown, some private sectors are affected and some of these sectors are closed which means that some individuals lost their jobs. Bianchi et al. [8] reported that in the USA, the unemployment rate rose from 3.8% in February 2020 to 14.7% in April 2020. Brookes et al. [10] reported that family financial loss can have even more problematic and enduring effects on children and adolescents.

Our results showed that 11-year-old children were less affected compared to all age groups during compared to pre-COVID-19; this difference was significant between 11 and 9-year-old children (*p* < 0.05). These findings are in disagreement with a previous study among children between 8–18 years old [12]. These authors reported that children’s age was not a moderating factor of the psychological distress of COVID-19. These authors argued that this might be attributed to the fact that their study does not include very young children. We think the older the child the better their thinking and understanding they have. Therefore, parents might include older children (11 years) in their discussions about financial and health consequences as a result of COVID-19, which decreases these psychological symptoms compared to young children. Dalton et al. (2019) indicated that children as young as 2 years old are aware of changes around them. Dalton et al. [24] also indicated even children younger than 2 years will notice the absence of regular caregivers (e.g., grandparents) and become unsettled and upset, seeking their return. Wang et al. [16] suggested that confinement could offer a good opportunity to enhance the interaction between parents and children, involve children in family activities, and improve their self-sufficiency skills. Perrin et al. [25] indicated that with the right parenting approaches, family bonds can be strengthened and child psychological needs met.

### 4.1. Study Implications

Our results have some implications for the development of psychological intervention programs to moderate the negative impact of COVID-19 on children and their families. In addition, Bull et al. [26] advised maintaining an adequate sleep rhythm and 60 min of moderate to vigorous daily PA to recover from these unwanted consequences of COVID-19. The official platform of the Ministry of Education in Jordan should provide classes about healthy lifestyles and psychosocial support programs available for schools and parents to choose from.

### 4.2. Study Strengths

This study has some strong points. First, it has a relatively big sample size (*n* = 1309 children). Second, it was the first study to assess the psychological impact of COVID-19 among children in Jordan. Last, it is one of the very few studies which have assessed, long-term (after one year of schools’ closure) the psychological impact of COVID-19 among children.

### 4.3. Limitation of the Study

This is a descriptive study. In addition, we did not correlate changes in children’s behaviour and emotional states to other variables such as parents’ stress. This was due to not making the questionnaire too long, especially when considering that parents filled in the survey.

## 5. Conclusions

School closure, lockdown, and social distancing are some measures implemented by the Jordanian government to reduce the health effect of COVID-19. These measures have affected children’s behaviour and emotional state during compared to before COVID-19. For example, children were more bored, irritable, and likely to argue with the rest of the family. In addition, screen time usage has also increased. Physical activity and sleeping hours have been decreased. The Jordanian Government needs to raise awareness of such psychological and mental health impacts of school closure during this unusual period, especially for children.

## Figures and Tables

**Table 1 ijerph-18-07795-t001:** Participant sociodemographic characteristics and mean score of parents perceived changes in children’s emotional and behavioural state.

	*n* (%)	Mean Score of Children’s Emotional and Behavioural State
**Parents gender**		
Male	90 (6.9)	3.10 ± 0.85
Female	1219 (93.1)	3.18 ± 0.73
**Living Place**		
City	1273 (97.2)	3.17 ± 0.73
Village	30 (2.3)	3.23 ± 0.63
Camp	2 (0.2)	2.61 ± 0.18
Bedouin	4 (0.3)	3.08 ± 1.63
**Marital Status**		
Married	1235 (94.3)	3.17 ± 0.74
Widowed	16 (1.2)	3.02 ± 0.78
Divorce	40 (3.1)	3.46 ± 0.57
Never married	18 (1.4)	2.93 ± 0.74
**Mostly used outside space?**		
Garden	456 (34.8)	3.14 ± 0.73
Balcony	666 (50.9)	3.19 ± 0.73
The roof of the building	131(10.0)	3.18 ± 0.74
Basement	56 (4.3)	3.21 ± 0.80
**Mother’s education**		
Less than High School	21(1.6)	3.18 ± 0.85
High School	114 (8.7)	3.16 ± 0.72
Diploma	192 (14.7)	3.08 ± 0.76
Bachelor	823 (62.9)	3.18 ± 0.72
Postgraduate	159 (12.1)	3.22 ± 0.76
**Father’s education**		
Less than High School	57 (4.4)	3.15 ± 0.81
High School	195 (14.9)	3.08 ± 0.71
Diploma	149 (11.4)	3.22 ± 0.74
Bachelor	690 (52.7)	3.18 ± 0.73
Postgraduate	218 (16.7)	3.19 ± 0.75
**Mother’s employment status**		
Self-employed	119 (9.1)	3.20 ± 0.78
Part time	90 (6.9)	3.20 ± 0.66
Full-time	403 (30.8)	3.18 ± 0.73
Unemployed	600 (45.8)	3.17 ± 0.73
Retired	30 (2.3)	2.83 ± 0.79
Lost job due to COVID-19	67 (5.1)	3.21 ± 0.71
**Father’s employment status**		
Self-employed	435 (33.2)	3.11 ± 0.77
Part time	70 (5.3)	3.16 ± 0.75
Full-time	638 (48.7)	3.20 ± 0.71
Unemployed	39 (3.0)	3.08 ± 0.75
Retired	56 (4.3)	3.02 ± 0.71
Lost job due to COVID-19	71 (5.4)	3.45 ± 0.68
**Children gender**		
Male	716 (54.7)	3.14 ± 0.72
Female	593 (45.3)	3.21 ± 0.75
**Education type**		
Private School	1213 (92.7)	3.17 ± 0.73
Public School	92 (7.0)	3.22 ± 0.78
Unrwa School	4 (0.3)	3.73 ± 0.92
Military culture School	0 (0)	
**School year group**		
Reception (5 yrs)	174 (13.3)	3.11 ± 0.78
Year 1 (6 yrs)	174 (13.3)	3.12 ± 0.76
Year 2 (7 yrs)	196 (15.0)	3.25 ± 0.72
Year 3 (8 yrs)	199 (15.2)	3.22 ± 0.74
Year 4 (9 yrs)	159 (12.1)	3.27 ± 0.65
Year 5 (10 yrs)	183 (14.0)	3.22 ± 0.70
Year 6 (11 yrs)	224 (17.1)	3.04 ± 0.74

Values are *n* (%), means, and standard deviation of the total score of parent’s perceived changes in children’s emotional and behavioural state.

**Table 2 ijerph-18-07795-t002:** Parents’ perceptions of the changes in child emotional and behavioural symptoms during COVID-19.

Child Symptoms	Much Less *n* (%)	Somewhat Less *n* (%)	Same *n* (%)	Somewhat More *n* (%)	Much More *n* (%)
My child is worried.	154 (11.8)	233 (17.8)	415 (31.7)	410 (31.3)	97 (7.4)
My child is restless.	167 (12.8)	212 (16.2)	293 (22.4)	495 (37.8)	142 (10.8)
My child is anxious.	169 (12.9)	218 (16.7)	336 (25.7)	449 (34.3)	137 (10.5)
My child is sad.	225 (17.2)	192 (14.7)	324 (24.8)	425 (32.5)	143 (10.9)
My child has nightmares.	332 (25.4)	227 (17.3)	538 (41.1)	178 (13.6)	34 (2.6)
My child is reluctant.	99 (7.6)	168 (12.8)	332 (25.4)	473 (36.1)	237 (18.1)
My child feels lonely.	194 (14.8)	183 (14.0)	245 (18.7)	447 (34.1)	240 (18.3)
My child is uneasy.	176 (13.4)	187 (14.3)	384 (29.3)	403 (30.8)	159 (12.1)
My child is nervous.	114 (8.7)	168 (12.8)	309 (23.6)	512 (39.1)	206 (15.7)
My child argues with the rest of the family.	77 (5.9)	123 (9.4)	314 (24.0)	536 (40.9)	259 (19.8)
My child is very quiet.	268 (20.5)	406 (31.0)	518 (39.6)	88 (6.7)	29 (2.2)
My child cries easily.	93 (7.1)	153 (11.7)	444 (33.9)	402 (30.7)	217 (16.6)
My child is angry.	113 (8.6)	165 (12.6)	353 (27.0)	465 (35.5)	213 (16.3)
My child feels frustrated.	180 (13.8)	177 (13.5)	394 (30.1)	408 (31.2)	150 (11.5)
My child is bored.	69 (5.3)	89 (6.8)	136 (10.4)	445 (34.0)	570 (43.5)
My child is irritable.	60 (4.6)	99 (7.6)	286 (21.8)	490 (37.4)	374 (28.6)
My child has no appetite.	321 (24.5)	252 (19.3)	556 (42.5)	124 (9.5)	56 (4.3)
My child has difficulty concentrating.	143 (10.9)	205 (15.7)	358 (27.3)	395 (30.2)	208 (15.9)
My child is afraid of COVID-19 infection.	192 (14.7)	177 (13.5)	435 (33.2)	318 (24.3)	187 (14.3)
My child is very dependent on us.	86 (6.6)	196 (15.0)	448 (34.2)	356 (27.2)	223 (17.0)
My child has behavioural problems.	257 (19.6)	217 (16.6)	503 (38.4)	238 (18.2)	94 (7.2)
My child eats a lot.	145 (11.1)	226 (17.3)	471 (36.0)	298 (22.8)	169 (12.9)
My child worries when one of us leaves the house.	147 (11.2)	188 (14.4)	568 (43.4)	275 (21.0)	131 (10.0)

**Table 3 ijerph-18-07795-t003:** Children’s patterns of daily screen time usage, daily physical activity, and sleep hours.

	*n* (%)
**Screen time usage (minutes)**	
Less than 30	79 (6.0)
From 30 to 60	187 (14.3)
From 60 to 90	229 (17.5)
From 90 to 120	174 (13.3)
From 120 to 180	189 (14.4)
More than 180	451 (34.5)
**Physical Activity (minutes)**	
Less than 30	550 (42.0)
From 30 to 60	355 (27.1)
From 60 to 90	174 (13.3)
From 90 to 120	96 (7.3)
From 120 to 180	66 (5.0)
More than 180	68 (5.2)
**Hours of sleep/day**	
Less than 6 h	43 (3.4)
7 h	93 (7.1)
8 h	419 (32.0)
9 h	300 (22.9)
10 h	306 (23.4)
11 h	78 (6.0)
More than 12 h	51 (3.9)

## Data Availability

Data are available from the authors (H.A. or R.B.) upon reason-able request.

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
