# Peer review of "Long-Term Psychological Effects of COVID-19 Pandemic on Children in Jordan"

_ijerph, 2021, doi:10.3390/ijerph18157795_

Round 1

Reviewer 1 Report

The article named “Long-term psychological effects of COVID-19 pandemic on children in Jordan” is proposed for publication with minor changes.

The authors propose a study on the negative consequences of COVID-19 on Jordanian children. They provide a good contextualization of the phenomenon, i.e., they address the literature on COVID-19 disease from its inception to the present day, both globally and in their particular context of study. They are particularly interested in the quarantine, school closures and social distancing measures applied and their consequences for children aged 5 to 11. They have a large sample size, as a total of 1309 parents participated in an online survey. This survey addressed questions aimed at measuring behavior and emotions of their children before and during the COVID-19 pandemic. I also consider this to be an innovative research, since, as the authors note, "there are no published studies in Jordan that have investigated the long-term impact of the COVID-19 pandemic on children's behaviour, emotional state and well-being".

The authors refer to their study purpose in line 72: "Therefore, this study was conducted to assess the effect of COVID-19 pandemic on psychological wellbeing among children between 5-11 years old in Jordan. We hypothesized that the COVID-19 will affect the psychological wellbeing of Jordanian children between 5-11 years old". However, I think it would be useful to specify clearly what the hypothesis of the study is/are. What are the specific hypothesis that the authors want to verify: will the COVID-19 pandemic affect the psychological wellbeing of Jordanian children positively, negatively, in all groups equally, do they think there will be differences by sex, age, family characteristics or status...?

Regarding the Participants section, it should be noted that the sample is quite homogeneous, which may to some extent impair the extrapolation of the data to the general population. On the other hand, this corresponds to greater precision when it comes to concluding and interpreting the results with respect to this specific sample. The high and higher participation of mothers (93.1%) compared to fathers (6.9%) is striking. The social status of the sample also emerges: 68.4% of the participants live in their own homes; 50.9% have balconies and 49.1% have access to an outdoor space for their children to play. In addition, 92.7% study in private schools. This confirms my view that the sample is highly homogeneous and with very specific characteristics and a high social and possibly economic status. With respect to boys and girls, the sample is gender and age balanced. It seems to me that Table 1 correctly collects the data and proposes a clear and visual picture of the data.

In the Procedures section, it is explained how the data collection was carried out. However, due to the high homogeneity of the sample, perhaps they should go a little deeper into this procedure. I wonder whether the way the questionnaire was published, as well as the social networks used or the idiosyncrasies of the interviewers may have played a role in, for example, the fact that 92.7% of the participants study in private schools. Perhaps the authors were targeting such population. I encourage clarification on this point.

As for the instrument used, it is well documented and relevant to the research objectives.

As far as data analysis is concerned, the authors propose correct analyses. However, I must express my misgivings about the samples used. For example: they have proposed an independent sample t-test to compare whether there was a difference in children's total behavioral and emotional states scores during, compared to before COVID-19, between children living in cities (1273 children) and those living in villages (30 children) and between those studying in private schools (1213 children) and in public schools (92 children). The large sample difference between the two groups does not give me confidence about the veracity of their comparisons. Perhaps the authors could review the need to compare these groups and change the focus of their research so that they focus only on the bulk of the sample: children living in towns (1273 children) and those studying in private schools (1213 children).

In terms of discussion and conclusions, the authors conclude that the measures taken by the Jordanian government to try to curb the COVID-19 pandemic have had negative consequences for children aged 5-11. School closures, blockades and social distancing have affected children's behavior and emotional states. For example, children were more bored, irritable and more likely to argue with the rest of the family. In addition, screen time has also increased. Physical activity and sleeping hours have decreased.

The authors also propose that the Jordanian government should raise awareness of these psychological and mental health impacts of school closures during this unusual period, especially for children. Under these headings they argue correctly and sufficiently for their interpretations and results, providing literature on the phenomenon to support their findings. However, I must reiterate my doubts about the extrapolation of the results to the general sample, due to the high homogeneity of the sample. I should also mention that, as the authors point out in the limitations of the study, the parents' own stress, difficulties and mental stability may be a determining factor in this study.

Finally, due to the content in the Discussion section, lines 243-257, on how stress, difficulties, etc., can affect parents and children wellbeing, I would find it interesting for future lines of research to include the variable resilience in the theoretical framework.

Author Response

We would like to thank the reviewer for such consrutuctive comments. 

The article named “Long-term psychological effects of COVID-19 pandemic on children in Jordan” is proposed for publication with minor changes.

The authors propose a study on the negative consequences of COVID-19 on Jordanian children. They provide a good contextualization of the phenomenon, i.e., they address the literature on COVID-19 disease from its inception to the present day, both globally and in their particular context of study. They are particularly interested in the quarantine, school closures and social distancing measures applied and their consequences for children aged 5 to 11. They have a large sample size, as a total of 1309 parents participated in an online survey. This survey addressed questions aimed at measuring behavior and emotions of their children before and during the COVID-19 pandemic. I also consider this to be an innovative research, since, as the authors note, "there are no published studies in Jordan that have investigated the long-term impact of the COVID-19 pandemic on children's behaviour, emotional state and well-being".

The authors refer to their study purpose in line 72: "Therefore, this study was conducted to assess the effect of COVID-19 pandemic on psychological wellbeing among children between 5-11 years old in Jordan. We hypothesized that the COVID-19 will affect the psychological wellbeing of Jordanian children between 5-11 years old". However, I think it would be useful to specify clearly what the hypothesis of the study is/are. What are the specific hypothesis that the authors want to verify: will the COVID-19 pandemic affect the psychological wellbeing of Jordanian children positively, negatively, in all groups equally, do they think there will be differences by sex, age, family characteristics or status...?

The first hypothesis was corrected by adding “negatively” and a new hypothesis on gender and social status and children’s age was added. As suggested by the reviewer.

We hypothesized that the COVID-19 will negatively affect the psychological wellbeing of Jordanian children between 5-11 years old. We also hypothesised that this negative effect of COVID-19 on the psychological wellbeing of Jordanian children will be affected by gender and social status of parents and children’s age. Finally, we hypothesised that physical activity level will be decreased and screen time usage will be increased among the study sample due to COVID-19.        

   Regarding the Participants section, it should be noted that the sample is quite homogeneous, which may to some extent impair the extrapolation of the data to the general population. On the other hand, this corresponds to greater precision when it comes to concluding and interpreting the results with respect to this specific sample. The high and higher participation of mothers (93.1%) compared to fathers (6.9%) is striking. The social status of the sample also emerges: 68.4% of the participants live in their own homes; 50.9% have balconies and 49.1% have access to an outdoor space for their children to play. In addition, 92.7% study in private schools. This confirms my view that the sample is highly homogeneous and with very specific characteristics and a high social and possibly economic status. With respect to boys and girls, the sample is gender and age balanced. It seems to me that Table 1 correctly collects the data and proposes a clear and visual picture of the data.

We would like to thank the reviewer for his supportive comment on table 1.

 In the Procedures section, it is explained how the data collection was carried out. However, due to the high homogeneity of the sample, perhaps they should go a little deeper into this procedure. I wonder whether the way the questionnaire was published, as well as the social networks used or the idiosyncrasies of the interviewers may have played a role in, for example, the fact that 92.7% of the participants study in private schools. Perhaps the authors were targeting such population. I encourage clarification on this point.

We did not target a specific group as 42% of Jordan population live in Jordan’s capital, Amman. This sentence was added in the procedures section to clarify it. The survey was distributed to parents’ pages and parents live in Jordan were asked to fill in the survey only.

 As for the instrument used, it is well documented and relevant to the research objectives.

 We would like to thank the reviewer for his supportive comment on this section.  

As far as data analysis is concerned, the authors propose correct analyses. However, I must express my misgivings about the samples used. For example: they have proposed an independent sample t-test to compare whether there was a difference in children's total behavioral and emotional states scores during, compared to before COVID-19, between children living in cities (1273 children) and those living in villages (30 children) and between those studying in private schools (1213 children) and in public schools (92 children). The large sample difference between the two groups does not give me confidence about the veracity of their comparisons. Perhaps the authors could review the need to compare these groups and change the focus of their research so that they focus only on the bulk of the sample: children living in towns (1273 children) and those studying in private schools (1213 children).

As suggested by the reviewer the analysis of t-test for school type (private and public) and living place (cities and villages) was removed. The focus of research then become on the total of the sample.   

In terms of discussion and conclusions, the authors conclude that the measures taken by the Jordanian government to try to curb the COVID-19 pandemic have had negative consequences for children aged 5-11. School closures, blockades and social distancing have affected children's behavior and emotional states. For example, children were more bored, irritable and more likely to argue with the rest of the family. In addition, screen time has also increased. Physical activity and sleeping hours have decreased.

 We would like to thank the reviewer for this supportive comment. 

The authors also propose that the Jordanian government should raise awareness of these psychological and mental health impacts of school closures during this unusual period, especially for children. Under these headings they argue correctly and sufficiently for their interpretations and results, providing literature on the phenomenon to support their findings. However, I must reiterate my doubts about the extrapolation of the results to the general sample, due to the high homogeneity of the sample. I should also mention that, as the authors point out in the limitations of the study, the parents' own stress, difficulties and mental stability may be a determining factor in this study.

 Thank you for such important comment. Although we did not measure parents’ own stress and difficulties in dealing with these measures such as school closure etc. and its correlation to children’s psychological wellbeing, similar and previous studies (Davico et al., 2021; Orgilés et al., 2020; Morgül et al., 2020) showed this correlation between parents’ own stress and children’s psychological wellbeing.    

Davico, C., Ghiggia, A., Marcotulli, D., Ricci, F., Amianto, F., & Vitiello, B. (2021). Psychological impact of the COVID-19 pandemic on adults and their children in Italy. Frontiers in psychiatry12, 239.

Orgilés, M., Morales, A., Delvecchio, E., Mazzeschi, C., & Espada, J. P. (2020). Immediate psychological effects of the COVID-19 quarantine in youth from Italy and Spain. Frontiers in psychology11, 2986.

Morgül, E., Kallitsoglou, A., & Essau, C. A. E. (2020). Psychological effects of the COVID-19 lockdown on children and families in the UK. Revista de Psicología Clínica con Niños y Adolescentes7(3), 42-48.

Finally, due to the content in the Discussion section, lines 243-257, on how stress, difficulties, etc., can affect parents and children wellbeing, I would find it interesting for future lines of research to include the variable resilience in the theoretical framework.

We would like to thank the reviewer for this supportive comment. 

Reviewer 2 Report

Thank you for the opportunity to review this manuscript. The researchers collected survey data from parents of youth (5-11 years) inquiring about children’s psychological, emotional, physical activity, sleep, and screen time behaviors during the Covid-19 pandemic. Results indicated that youth slept less, engaged in less physical activity, engaged in more screen time, and felt more bored, lonely, and irritable than prior to the pandemic. Strengths of this study include the sample size, unique sample (i.e., Jordan families), and relevance to Covid-19. This reviewer has two major concerns about the paper. First, the use of the term “long-term psychological effects” is never defined in the paper. Given that the data includes a one-time parent self-report of children’s perceived emotions and behaviors, I do not believe this is consistent with “long-term psychological effects.” Second, the discussion/conclusion sections need significant re-working. There is a general lack of clarity and organization, as well as many assumptions made based upon study results that have no (or limited) evidence to support them. I caution the authors to be careful of using language that is causal based upon cross-sectional data, and I encourage the use of “perceived psychological and emotional symptoms” since these symptoms are parent-reported. Suggested edits and other concerns are listed below in more detail.

General concerns:

  1. Keep tenses consistent throughout sections
  2. Needs editing/grammar checking. There are a number of typos.
  3. I suggest you change the phrase “time of screen use” to “screen time duration” or “screen time usage” (or something similar). This is more consistent with the literature in this area.

Introduction:

  1. Could use some reworking. The first 3 paragraphs are interesting, but seem too detailed on the pandemic and resulting infrastructure issues. For example, in paragraph two the authors discuss prices of agricultural commodities and shortages in N95 face masks. How does this relate to your study on exercise, screen time, and youth’s boredom/loneliness?
  2. You mention in lines 59-64 how parent stress can be associated with children’s wellbeing. This makes perfect sense and is well-stated, however you do not measure parents stress at all in the paper. I might remove as it is tangential to the paper.
  3. The introduction would benefit from further justification for the purpose of the paper and what potential the results can offer. More information on youth emotions and health behaviors is needed.
  4. Lines 74-75. Recommend including directionality in your hypothesis (i.e., did you expect the pandemic to positive or negatively affect the psychological wellbeing of children)?
  5. Add hypothesis about sleep, activity, and screen time.

Materials and Methods:

  1. How did the researchers ensure that only parents from Jordan completed the survey, as the social networks they used to distribute the survey are used internationally?
  2. Were participants compensated for their time?
  3. Please explain further how the survey was first shared via social media, for example was it shared to parenting groups/pages or via the researchers’ personal profiles?
  4. Lines 125-127: How did you compare emotional state differences from before to during the pandemic? It is unclear as the questionnaire describes a 5-pt Likert scale. Did the authors combine and compare for example 1-2 to 4-5 answers? Please clarify.
  5. There doesn’t seem to be enough participants who reported living in villages to compare to those who live in cities. How can you be sure that the results are meaningful/generalizable given your sample?

Results:

  1. Lines 163-164- see comment #12.
  2. Lines 165-166: Do not need to list all of the age categories as it’s listed in the table.

Discussion:

  1. Lines 202-204: This sentence seems to come out of nowhere for me, particularly in discussing teachers. Either improve the transition or consider moving to a clinical implications section.
  2. Lines 205-209: This is a great connection to children’s weekend and summer holiday behaviors; however the authors need to more clearly articulate that they are comparing the pandemic behaviors to these.
  3. Line 207: Is this sentence complete? (“… and summer holidays they.”)
  4. Lines 207-213: Please tighten up the language here to make it clearer what you are stating. Odd to state that the lockdown “will limit activities…” when you are discussing something from the past. What suggestions?…
  5. Remove parenthesis in line 214 and 218 around %s.
  6. Line 22: Typo “doing”
  7. Lines 228-233: Very good hypothesis, however not sure you can state the causality as strongly given your results. Adjust wording.
  8. Lines 237-240: The stated changes in sleep time from Abid et al. do not seem very significant. Did the authors from that paper state any associations with these changes that may be relevant to your paper? May be helpful to include. Additionally, is there a way to more directly compare your results to the Abid et al results? As it’s written, comparing 42% of youth with less than 8 hours/night is a less direct comparison to “reduction by 0.07 hours…”. Consider changing or adding more information (i.e., reduction from what mean?).
  9. Lines 245-250: Consider adding that caregivers may have spent less time with caregivers if children of divorced families were unable to travel from one parent’s residence to another during the lockdown, or if this transfer occurred less frequently.
  10. Lines 250-254: Not all readers will be familiar with the CRIES-13. Remove the specific measure and replace with the construct it measures.
  11. Lines 254-259: Edit- this sentence is a run-on.
  12. Last paragraph in the discussion: what other child developmental considerations may explain the results?
  13. Lines 274-278: Assuming causality/explaining findings without sufficient evidence with “this confirms that parents will evolve….” Re-phrase.
  14. The Discussion section would benefit from a future directions and/or clinical implications section as well as a strengths/limitations section (see below).

Conclusions:

  1. See comment #29. Move second half of this paragraph to clinical implications/future directions. Generally, would not recommend adding new content to the Conclusion section.
  2. Consider moving study strengths/limitations to the Discussion section.

Tables:

  1. Table 1: Remove Total row under each section. It doesn’t add anything meaningful.
  2. Table 2: While I find this data very interesting, I’m not sure you need to include every question asked in the survey as the authors do not discuss all the results. Fine to leave as is, however I might consider removing some of the rows not discussed.
  3. Table 2: I am curious why the authors didn’t mention the two questions that appear to have declined over the pandemic: “my child is very quiet” and “my child has no appetite.” Particularly the latter has potentially interesting results for child eating behaviors which aligns nicely with the other behaviors discussed in this paper.
  4. Table 3: The font is off in the table, such that the first N result under each section appears much larger than the rest of the results.

Author Response

Reviewer 2#

General concerns:

#Keep tenses consistent throughout sections

The paper was gone through English proofreading after making the amendments.

#Needs editing/grammar checking. There are a number of typos.

Please see comment 1#

#I suggest you change the phrase “time of screen use” to “screen time duration” or “screen time usage” (or something similar). This is more consistent with the literature in this area.

We thank the reviewer for this comment. The phrase “time of screen use” was replaced by “screen time usage” throughout the paper. 

Introduction:

#Could use some reworking. The first 3 paragraphs are interesting, but seem too detailed on the pandemic and resulting infrastructure issues. For example, in paragraph two the authors discuss prices of agricultural commodities and shortages in N95 face masks. How does this relate to your study on exercise, screen time, and youth’s boredom/loneliness?

In Jordan, private sectors decreased the salaries between 25-30% which affected parent’s purchasing power. This will in turn affect the psychological wellbeing of parents and children. In addition, some sectors are closed. Therefore, children of parents who lost their jobs as a result of COVID 19 were more affected than who did not lose their jobs. In addition, in March 2021 there was low numbers of ICU beds and ventilators due the high number of new cases of COVID 19 in Jordan (each day there were more than 10 thousands cases a day).  Moreover, the media announced on summer 2020 that some parents transfer their children from private to public schools due to school closure due to COVID 19. These things have affected parents psychological well being and children.    

#You mention in lines 59-64 how parent stress can be associated with children’s wellbeing. This makes perfect sense and is well-stated, however you do not measure parents stress at all in the paper. I might remove as it is tangential to the paper.

Thank you for such important comment. Although we did not measure parents’ own stress and difficulties in dealing with these measures such as school closure etc. and its correlation to children’s psychological wellbeing in our study, similar and previous studies (Davico et al., 2021; Orgilés et al., 2020; Morgül et al., 2020) showed this correlation between parents’ own stress and children’s psychological wellbeing.   

Davico, C., Ghiggia, A., Marcotulli, D., Ricci, F., Amianto, F., & Vitiello, B. (2021). Psychological impact of the COVID-19 pandemic on adults and their children in Italy. Frontiers in psychiatry12, 239.

Orgilés, M., Morales, A., Delvecchio, E., Mazzeschi, C., & Espada, J. P. (2020). Immediate psychological effects of the COVID-19 quarantine in youth from Italy and Spain. Frontiers in psychology11, 2986.

Morgül, E., Kallitsoglou, A., & Essau, C. A. E. (2020). Psychological effects of the COVID-19 lockdown on children and families in the UK. Revista de Psicología Clínica con Niños y Adolescentes7(3), 42-48.

So we would like to keep it as it is please.

#The introduction would benefit from further justification for the purpose of the paper and what potential the results can offer. More information on youth emotions and health behaviors is needed.

Please see paragraph number 4 in the introduction. This paragraph talks about children’s emotions and the impact of COVID 19 on it. Specifically this sentence was added “Children and adolescents may be more susceptible because of home confinement, school closure, lack of in-person contact with classmates, friends and teachers, and limitation in personal space at home [10; 11; 12; 13; 14]”.   

These two sentences were added before the hypotheses and after the aim of the study to clarify the importance of such findings.  

This study will have important implications for ministries of education and health in Jordan by adding some lectures on how parents can manage their children’s stress and mental health during such difficult times. In addition, these ministries should provide more lectures on the importance of increasing physical activity and less screen time usage for children.  

#Lines 74-75. Recommend including directionality in your hypothesis (i.e., did you expect the pandemic to positive or negatively affect the psychological wellbeing of children)?

It was corrected. We hypothesized that the COVID-19 will negatively affect the psychological wellbeing of Jordanian children between 5-11 years old.

# Add hypothesis about sleep, activity, and screen time.

A hypothesis was added as suggested by the reviewer. Finally, we hypothesised that physical activity level will be decreased and screen time usage will be increased among the study sample due to COVID-19.        

Materials and Methods:

#How did the researchers ensure that only parents from Jordan completed the survey, as the social networks they used to distribute the survey are used internationally?

This sentence was added in the procedures section to clarify this. The survey was distributed to parents’ pages and parents live in Jordan were asked to fill-in the survey only.

#Were participants compensated for their time?

It was voluntary. We referred to that in the acknowledgment section by thanking the parents who filled in the survey. “Special thanks also to everyone who participated in our survey for sharing their views and personal experiences during challenging times of COVID19 especially when considering that it was voluntary”.  This sentence was also added to the procedures section. Parents were also informed that they will not be paid for filling in the survey.   

#Please explain further how the survey was first shared via social media, for example was it shared to parenting groups/pages or via the researchers’ personal profiles?

We did not target a specific group as 42% of Jordan population live in Jordan’s capital, Amman. This sentence was added in the procedures section to clarify it. The survey was distributed to parents’ pages and parents live in Jordan were asked to fill in the survey only.

#Lines 125-127: How did you compare emotional state differences from before to during the pandemic? It is unclear as the questionnaire describes a 5-pt Likert scale. Did the authors combine and compare for example 1-2 to 4-5 answers? Please clarify.

These two sentences were added at the beginning of the result’s section to clarify this point.

In order to assess parents’ perceived changes in children’s behaviour and emotional states during compared to before the COVID-19, the percentages of somewhat more and much more were summed. For example, 31.3% of parents reported the changes in “my child is worried” as somewhat more and 7.4% of them reported these changes as much more, this gives the perceived changes as 38.7%.

#There doesn’t seem to be enough participants who reported living in villages to compare to those who live in cities. How can you be sure that the results are meaningful/generalizable given your sample?

We would like to thank the reviewer for this comment. As suggested by the reviewer the analysis of t-test for school type (private and public) and living place (cities and villages) was removed. The focus of research then become on the total of the sample.   

Results:

#Lines 163-164- see comment #12.

In the footnote of table 1, it is written that the values of the total score of parent’s perceived changes in children’s emotional and behavioural state total score are means ± SD. For more clarification, we summed the scores of parent’s perceived changes in children’s emotional and behavioural state of all items and divided it by the number of items.   

#Lines 165-166: Do not need to list all of the age categories as it’s listed in the table.

Age categories are clearly mentioned in table 1. Therefore, it was removed from text as suggested.

Discussion:

#Lines 202-204: This sentence seems to come out of nowhere for me, particularly in discussing teachers. Either improve the transition or consider moving to a clinical implications section.

Thank you for such nice comment. Transition was considered as suggested.

This can be attributed to the fact that teachers have……

#Lines 205-209: This is a great connection to children’s weekend and summer holiday behaviors; however the authors need to more clearly articulate that they are comparing the pandemic behaviors to these.

Thanks a lot for this comment. In fact, one of the references talks about weekend and summer holiday and the second one discuss physical activity and screen time usage during the lockdown due to COVID19.

Moreover, Wang et al. [16] reported that lockdown due to COVID-19 restricted outdoor activities and interaction with same-aged friends during the outbreak among children which in turn will worsen such negative effects.

#16. Wang, G., Zhang, Y., Zhao, J., Zhang, J., & Jiang, F. (2020). Mitigate the effects of home confinement on children during the COVID-19 outbreak. The Lancet, 395(10228), 945-947.

#Line 207: Is this sentence complete? (“… and summer holidays they.”)

The pronoun they was deleted. The sentence now is complete.  

#Lines 207-213: Please tighten up the language here to make it clearer what you are stating. Odd to state that the lockdown “will limit activities…” when you are discussing something from the past. What suggestions?…

Thanks a lot for this comment. It is re-phrased now.

 Wang et al. [16] reported that lockdown due to COVID-19 restricted………….

These perceived changes in physical activity, screen time usage and sleeping hours observed by Spinelli et al. [17]…………

#Remove parenthesis in line 214 and 218 around %s.

Parentheses were removed as suggested.  

#Line 22: Typo “doing”

The typo error was corrected, in doing less than 30 minutes …….

#Lines 228-233: Very good hypothesis, however not sure you can state the causality as strongly given your results. Adjust wording.#

The causality was removed. Therefore, school closure and full and partial lockdown due to COVID-19 led to less physical activity, more screen time usage, irregular sleep patterns, and less favourable diets.

#Lines 237-240: The stated changes in sleep time from Abid et al. do not seem very significant. Did the authors from that paper state any associations with these changes that may be relevant to your paper? May be helpful to include. Additionally, is there a way to more directly compare your results to the Abid et al results? As it’s written, comparing 42% of youth with less than 8 hours/night is a less direct comparison to “reduction by 0.07 hours…”. Consider changing or adding more information (i.e., reduction from what mean?).

The mean of sleeping hours in Abid et al. (2021) study were added for both boys and girls to clarify the reduction from the mean for boys and the increase in the mean for girls.

They reported a reduction in sleeping hours among boys by 0.07 hours during (8.71±0.93 h) compared to before lockdown (8.78 ±0.95 h) and an increase in sleeping hours among girls by 0.08 hours during (8.73±0.78 h) compared to before lockdown (8.65 ±0.72 h).

#Lines 245-250: Consider adding that caregivers may have spent less time with caregivers if children of divorced families were unable to travel from one parent’s residence to another during the lockdown, or if this transfer occurred less frequently.

This statement was added, as suggested by the reviewer, a possible interpretation why children of divorced families are more affected than children of married families.  

This may also be attributed to the fact that children of divorced families may have spent less time with caregivers if children were unable to travel from one parent’s residence to another during the lockdown, or if this transfer occurred less frequently.    

#Lines 250-254: Not all readers will be familiar with the CRIES-13. Remove the specific measure and replace with the construct it measures.

It was replaced by this sentence “children’s post-traumatic stress scores were related to their parents’ post-traumatic stress symptom scores……

#Lines 254-259: Edit- this sentence is a run-on.

The sentence is re-phrased as suggested.

Maiti et al. [23] reported that the strength of a good marital or couple bonding can actually make both the individuals internally strong and confident, which definitely help to fight this distress in a better way, individually, and also as a strong united couple. Therefore, divorced couple usually do not have good marital relationship which is going to be reflected negatively on their children.

#Last paragraph in the discussion: what other child developmental considerations may explain the results?

The discussion of this point was re-phrased to emphasise that older children (11 years old) were less affected than younger children (10 and younger).  Our results showed that 11 years old children were less affected compared to all age groups during compared to pre-COVID-19; this difference was significant between 11 and 9 years old children (P < 0.05).  These findings are in disagreement with previous study among children between 8-18 years old [12]. These authors reported that children’s age was not a moderator factor of psychological distress of COVID-19. These authors argued that this might be attributed to the fact that their study does not include very young children.  We think the older the child the better their thinking and understanding he has.

#Lines 274-278: Assuming causality/explaining findings without sufficient evidence with “this confirms that parents will evolve….” Re-phrase.

The sentence was re-phrased. Therefore, parents might evolve older children …………..

#The Discussion section would benefit from a future directions and/or clinical implications section as well as a strengths/limitations section (see below).

The second half of the paragraph in the conclusion was moved to the discussion. 

Conclusions:

#See comment #29. Move second half of this paragraph to clinical implications/future directions. Generally, would not recommend adding new content to the Conclusion section.

Yes, the second half of paragraph was moved to the discussion.

Our results have some implications for the development of psychological intervention programs to moderate the negative impact of COVID-19 on children and their families. In addition, Bull et al. [26] advised maintaining an adequate sleep rhythm and 60 min of moderate to vigorous daily PA to recover these unwanted consequences of COVID 19. The official platform of the Ministry of Education in Jordan should provide classes about healthy lifestyles and psychosocial support programs available for schools and parents to choose from. 

#Consider moving study strengths/limitations to the Discussion section.

Study strengths and limitations were moved to the discussion section.

Tables:

#Table 1: Remove Total row under each section. It doesn’t add anything meaningful.

Total row was removed from table 1 as suggested.

#Table 2: While I find this data very interesting, I’m not sure you need to include every question asked in the survey as the authors do not discuss all the results. Fine to leave as is, however I might consider removing some of the rows not discussed.

Although we did not discuss all the questions, all of them were analyzed. We would like to leave as it is, as some data may be compared with future studies.  

#Table 2: I am curious why the authors didn’t mention the two questions that appear to have declined over the pandemic: “my child is very quiet” and “my child has no appetite.” Particularly the latter has potentially interesting results for child eating behaviors which aligns nicely with the other behaviors discussed in this paper.

Yes, these two questions were added in the result section and “my child has no appetite” was added in the discussion. In addition, 51.5% of parents perceived their children to be less quiet and 43.8% of parents reported that their children have no appetite.     

#Table 3: The font is off in the table, such that the first N result under each section appears much larger than the rest of the results.

#The font in table 3 was checked. The font in table 1, 2 and 3 is 10 Palatino Linotype.   

Round 2

Reviewer 2 Report

None. Thank you for thoughtfully addressing all concerns.